# NV⁻ diamond laser

Alexander Savvin[1], Alexander Dormidonov [1✉], Evgeniya Smetanina[1], Vladimir Mitrokhin[1], Evgeniy Lipatov[2,3], Dmitriy Genin[2,3], Sergey Potanin [4,5], Alexander Yelisseyev[6] & Viktor Vins[7]

For the first time, lasing at NV⁻ centers in an optically pumped diamond sample is achieved. A nanosecond train of 150-ps 532-nm laser pulses was used to pump the sample. The lasing pulses have central wavelength at 720 nm with a spectrum width of 20 nm, 1-ns duration and total energy around 10 nJ. In a pump-probe scheme, we investigate lasing conditions and gain saturation due to NV⁻ ionization and NV⁰ concentration growth under high-power laser pulse pumping of diamond crystal.

[1] Dukhov Automatics Research Institute, Sushchevskaya 22, Moscow 127055, Russian Federation. [2] National Research Tomsk State University, 36 Lenin ave., Tomsk 634050, Russian Federation. [3] Institute of High Current Electronics SB RAS, 2/3 Akademicheskii ave., Tomsk 634055, Russian Federation. [4] Faculty of Physics, Lomonosov Moscow State University, Leninskie Gory 1, Moscow 119991, Russian Federation. [5] Sternberg Astronomical Institute, Lomonosov Moscow State University, Universitetskii ave. 13, Moscow 119234, Russian Federation. [6] V.S. Sobolev Institute of Geology and Mineralogy SB RAS, 3 Koptyug ave., Novosibirsk 630090, Russian Federation. [7] LLC VELMAN 43, Russkaya str, Novosibirsk 630058, Russian Federation. ✉email: dormidonov@gmail.com

Color centers in diamond exhibit outstanding spectroscopic properties[1] with photoluminescence spectra covering visible and near-IR wavelength ranges[2–6]. Optical transparency of diamond and its excellent mechanical, thermal, and optical characteristics make it a promising host material for color-center photon source[7,8]. One of the most studied color centers in diamond, the negatively charged nitrogen-vacancy (NV⁻) center, is a unique solid-state quantum system exhibiting high quantum efficiency at room temperature. It can be coupled to dielectric cavities and plasmonic resonators in order to create single-photon emitters applicable in quantum photonic technologies[9–11]. Laser initialization, changing and reading the spin state of an NV⁻ electron allows to control the state of a quantum system and turn NV⁻ centers in diamond into a promising platform for optical quantum computing[12–19], quantum metrology, sensing and visualization, fluorescent magnetometry and laser threshold magnetometry that requires a CW laser constructed from the NV⁻ centers[20–26].

For decades, diamond crystals with nitrogen-vacancy centers have been considered as a potential lasing medium[2,27,28]. Nitrogen vacancy color centers exist in two states: negatively charged—NV⁻ (with its zero-phonon line at 637 nm) and neutral—NV⁰ (with its zero-phonon line at 575 nm)[1]. Both states support strong electron–phonon interactions that result in a very broad phonon sideband (PSB) emission[1,11]. Exhibiting large emission and absorption cross sections, high quantum efficiency and a broad gain band, diamond with NV centers can be considered as a promising solid state active media for tunable and femtosecond lasers in the red and near infrared spectral regions[29].

Monocrystalline samples of IIa-type diamond with NV color centers created by means of high pressure and high temperature synthesis (HPHT) are currently available on the market[29–31]. Spectroscopic characteristics of NV⁻ centers in diamond were investigated in[27]; but, stimulated emission or lasing at NV centers in diamond have not been achieved in that work. Stimulated emission at wavelengths in the PSB of NV⁻ zero-phonon line (ZPL) was registered in[28] through lock-in detection consistent with a simultaneous reduction in spontaneous emission under a CW optical pump at 532 nm. In the recent work[32] the amplification of 721-nm laser radiation by stimulated emission from NV⁻ centers in diamond inside an optical fiber cavity was demonstrated. However, a lasing at NV⁻ centers in diamond have not been realized yet.

In this work, we investigated in detail spectral properties of the HPHT diamond sample with two growth zones exhibiting different NV⁰ to NV⁻ concentration ratios. Sample photoluminescence was enhanced by CW and pulsed lasers at various wavelengths. A direct pump-probe method was used for registration of stimulated emission from the diamond sample. We demonstrate that the diamond sample zone with a predominance of NV⁰ centers exhibits only pump-induced absorption of the probe, while the growth zone with a high concentration of NV⁻ centers provides the probe amplification. However, we detect a saturation of the probe amplification and probe absorption that appears due to NV⁻ ionization and NV⁰ formation at sufficiently high pump rates. The maximal detected gain coefficient is about 1.5 cm⁻¹ that allows us to reach lasing conditions in the diamond sample zone with a predominant NV⁻ concentration and, for the first time, we achieve a laser generation at NV⁻ color centers in diamond.

## Results

### Spectroscopic characterization of the diamond sample.

An investigated diamond sample is a $4 \times 3 \times 0.25$ mm plate containing two growth zones (Fig. 1) with significantly different NV⁰ to NV⁻

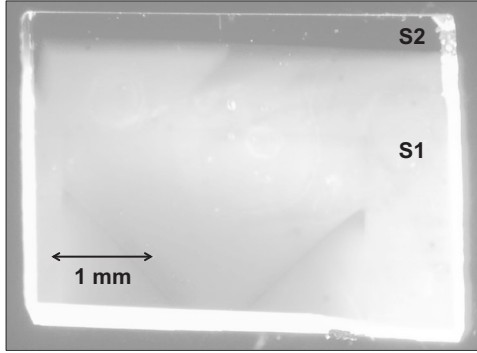

**Fig. 1 The diamond sample.** Photography of the diamond sample that has two growth zones with significantly different ratios of NV⁰ concentration to NV⁻ concentration. Zone S1 mainly contains neutral NV⁰ centers and NV⁻ concentration dominates in zone S2.

concentration ratios. The largest part of the sample (zone S1) mostly contains neutral NV⁰ centers. In a narrow (about 250 μm) dark-colored region at the edge of the sample (zone S2) the concentration of NV⁻ centers turned out to be significantly higher than the NV⁰ concentration. Two lateral faces of the sample were polished at the Brewster's angle (See Methods for more details).

The absorption spectra after Fresnel correction are given in Fig. 2a in the units of cm⁻¹ (gray curve for S1 and black curve for S2 sample zones, respectively). The absorption coefficient $\alpha(\lambda) = \frac{1}{d} \ln \frac{I_0(\lambda)}{I(\lambda)}$ was obtained by illuminating the sample by a tungsten incandescent lamp with incident intensity $I_0(\lambda)$ and transmitted intensity $I(\lambda)$ through the sample with thickness $d = 0.25$ mm for each zone. Transmitted intensity $I(\lambda)$ was measured at room temperature employing an Ocean Optics FLAME-S-XR1 spectrometer with optical resolution 1.69 nm FWHM and 200-μm fiber input for light collection. The obtained optical absorption spectra are typical for type IIa diamonds[1,6].

The S1-zone absorption spectrum has a maximum at 575 nm and a width of about 150 nm (Fig. 2a, gray curve), which corresponds to the characteristic absorption spectrum of NV⁰ centers in diamond[27,28]. The S2-zone absorption spectrum contains a pronounced absorption band from 500 to 660 nm with a maximum around 569 nm due to the domination of NV⁻ centers in this zone (Fig. 2a, black curve). At the peak NV⁻ absorption wavelength the absorption coefficient reaches 14 cm⁻¹, which is an order of magnitude less than the NV⁻-containing diamond sample absorption coefficient measured in[27]. We observed strong blue absorption in the spectral range from 400 to 495 nm (Fig. 2a, black curve) that indicated a significant amount of nitrogen atoms in the S2 zone[33]. The nitrogen concentration $3.1 \times 10^{19}$ cm⁻³ (175 ppm) in S2 zone was measured employing Bruker Vertex 70 FTIR spectrometer combined with a micro-scope Hyperion 2000. Electrons of nitrogen donor atoms (C-centers) are captured by NV⁰ centers, thereby changing their state to NV⁻[34]. Thus, the presence of nitrogen atoms in the S2 zone naturally leads to an increase in the NV⁻ concentration and, we believe, is the key difference between the zones S1 and S2.

We used three CW lasers with a power of about 10 mW at wavelengths of 400, 450, and 532 nm to illuminate the sample. The sample was mounted on a vertical translation stage with a micrometric step resolution allowing to operate with sample zones S1 and S2 without any readjustment of the experimental set-up. The photoluminescence spectra were collected from the illuminated sample surface, i.e. the incident laser beam and the recording spectrometer were located on the same side of the

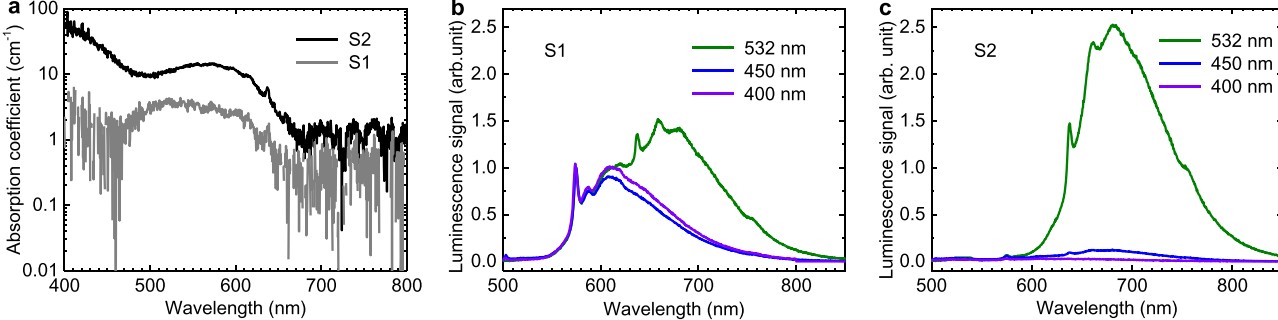

**Fig. 2 Spectroscopic characterization of the diamond sample. a** Absorption spectrum of S1 (gray curve) and S2 (black curve) zones of diamond sample. The spectra were recorded at room temperature. Normalized photoluminescence spectra of S1 (**b**) and S2 (**c**) zones under continuous excitation at various wavelengths: 532 nm (green curve), 450 nm (blue curve), and 400 nm (violet curve). For each pump wavelength the spectra collected from both zones were normalized on the S1-zone spectral amplitude at 575 nm that corresponds to the $NV^0$ zero-phonon line. The photoluminescence spectra were collected from the sample excitation surface, i.e. the pump lasers and the recording spectrometer were located on the same side of the sample.

sample. Normalized photoluminescence spectra of S1 and S2 zones recorded at room temperature employing the Ocean Optics FLAME-S-XR1 spectrometer are given in Fig. 2b and c, respectively. For each excitation wavelength, the spectra collected from both zones were normalized on the S1-zone spectral amplitude at 575 nm that corresponds to the $NV^0$ ZPL in diamond. The photoluminescence spectra of the zone S1 exhibit a strong $NV^0$ centers ZPL at 575 nm and its PSB emission with a maximum at 609 nm at all considered excitation wavelengths (Fig. 2b). ZPL of $NV^-$ centers at 638 nm and its PSB emission with a maximum at 680 nm are presented in the photoluminescence spectra of both zones under 532-nm excitation (Fig. 2b, c green curves). The photoluminescence spectra under 400- and 450-nm excitation for zone S1 are identical (Fig. 2b blue and violet curves) and have much higher amplitudes than the photoluminescence spectra from S2 zone (Fig. 2c blue and purple curves) because both excitation wavelengths fall in the $NV^0$ absorption region and are not absorbed by the $NV^-$ centers.

The decay time of the spontaneous emission signal integrated over in the wavelength range 670–680 nm was found to be $\tau_{21} = 9.3 \pm 0.3$ ns for the S2 zone with a predominant concentration of $NV^-$ color centers. We employed a fast optoelectronic converter to measure $\tau_{21}$ (See Methods for more details). The obtained decay time is close to the $NV^-$ excited state lifetime given in a number of other works: $8 \pm 1$ ns in[35], 7.8 or 12 ns depending on the spin state of the diamond defect[36], 12 and 13 ns in[27,28] and[29], respectively.

Based on photoluminescence spectra of the sample zone S2 we calculate emission cross section $\sigma_{em}$ at the center of the emission band $\lambda_{em} = 680$ nm (See Methods for details). The estimated value reaches $\sigma_{em} = 4.9 \cdot 10^{-17}$ cm$^2$ that is close to the emission cross-section values $4.3 \cdot 10^{-17}$ cm$^2$ and $3.6 \cdot 10^{-17}$ cm$^2$ reported in[27] and[35], respectively, and it is one order below the value $3.2 \cdot 10^{-16}$ cm$^2$ given in[29]. Corresponding absorption cross section at the center of the absorption band $\lambda_{abs} = 569$ nm of zone S2 (See Methods for details) is $\sigma_{abs} = 5.7 \cdot 10^{-17}$ cm$^2$ that matches well with the absorption cross-section values reported in following works: $2.8 \cdot 10^{-17}$ cm$^2$ [27], $(3.1 \pm 0.8) \cdot 10^{-17}$ cm$^2$ [37], and $(9.5 \pm 2.5) \cdot 10^{-17}$ cm$^2$ [38]. Using the obtained absorption cross-section $\sigma_{abs}$ and corresponding absorption coefficient $\alpha$ from Fig. 2a, the concentration $N_{NV^-} = \frac{\alpha}{\sigma_{abs} d}$ of $NV^-$ centers is found to be ~$2.5 \cdot 10^{17}$ cm$^{-3}$ (1.4 ppm) in the S2 zone. This value lies in the range of the $NV^-$ concentrations corresponding to the lasing conditions predicted in[28]. Note that in work[27], where lasing was not achieved, the concentration of $NV^-$ color centers was

$4.5 \cdot 10^{18}$ cm$^{-3}$, which is an order of magnitude higher than the concentration of $NV^-$ centers in the S2 zone of our sample.

**Pump-saturation effect of photoluminescence.** To investigate a photoluminescence at high intensity pump rate, we used a pico-second Nd:YAG laser designed in our laboratory that delivers a train of 150-ps pulses delayed by 4.4 ns from each other with possibility of a high contrast selection of a single pulse[39]. Total energy of the 532-nm pulses train has reached 400 µJ and could be attenuated by set of filters introduced in the optical path. The average light field intensity of pulses train focused into a spot with a diameter of 120 µm (FWHM) reaches about $5 \cdot 10^7$ W/cm$^2$, and the peak intensity of single pulse is $3 \cdot 10^9$ W/cm$^2$.

We have detected experimentally that for both S1 and S2 sample zones the spectral amplitude of $NV^0$ ZPL at 575 nm grows under pump energy increase (Fig. 3a and b, respectively). In the zone S1 the shape of the luminescence spectrum has characteristic features defined by $NV^0$ PSB for all considered incident pulse train energies, while $NV^-$ ZPL at 638 nm and its PSB are weakly pronounced. In the zone S2 $NV^-$ PSB defines the luminescence spectrum shape, which has a maximum at ~680 nm for low pump energies (Fig. 3b, black curve). The pump energy increase from 0.1 µJ up to 400 µJ leads to the enhancement of the $NV^-$ centers photo-ionization process and corresponding $NV^0$ concentration growth in the S2 zone. As a result, we observe a saturation of the $NV^-$ contribution and appearance of the $NV^0$ ZPL and PSB in the range of 575–630 nm in the photoluminescence spectrum (Fig. 3b, blue and red curves). The observed pump-saturation effect of $NV^-$ photoluminescence is in a good agreement with the works[27,40–42], where the role of the $NV^-$ photo-ionization process under pump power increase was investigated.

**Stimulated emission from $NV^-$ centers in diamond.** We employed a direct pump-probe method for stimulated emission registration from the S1 and S2 zones of the diamond sample (Fig. 4). A 675-nm Thorlabs HL6750MG CW laser diode with 100-mW power was used to probe the sample. In the first series of experiments, the sample was placed perpendicularly to the probe beam at a 150-mm distance from the laser diode. An 11-mm lens creates a 50-µm (FWHM) probe beam waist at the sample surface (Fig. 4, inset 1). A vertical translator with a micrometer resolution provided an opportunity to switch between the studied sample zones S1 and S2. In this case, to pump the sample in a single-pulse regime, we cut the central laser pulse from a pulse train of 150-ps pulses[39]. The energy of the 532-nm single pulse was controlled by attenuating filters in the range from 0.6 to 15 µJ.

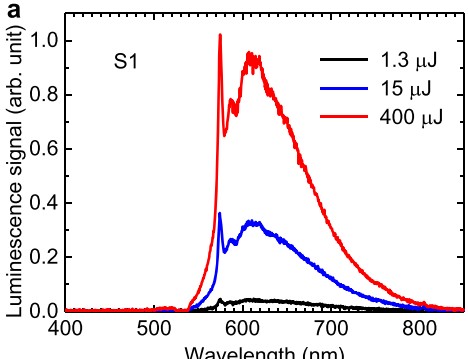
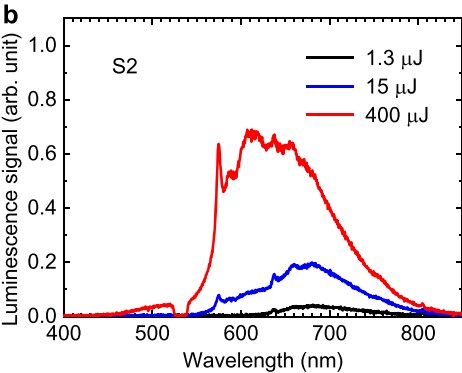

**Fig. 3 Pump-saturation effect of photoluminescence.** Photoluminescence spectra of (**a**) S1 and (**b**) S2 sample zones under excitation of 532-nm pulsed pump with the pulse train total energy of 1.3 μJ (black curve), 15 μJ (blue curve), 400 μJ (red curve). The dip in the spectra around the pump wavelength of 532 nm is related to the Thorlabs NF533-17 Notch filter present in the scheme. The spectra were recorded in transmission configuration; the pump laser and the recording spectrometer were located on opposite sides of the sample.

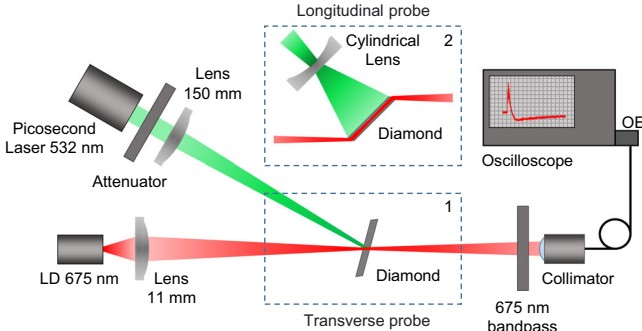

**Fig. 4 Experimental setup.** Scheme of a pump-probe experiment for registration of stimulated emission from the diamond sample. We employed transverse probe configuration in which the sample was placed perpendicularly to the probe beam (inset 1) and longitudinal probe configuration in which probe beam was going along the diamond parallel to sample side surfaces (inset 2).

The waist diameter of the pump beam focused by a 150-mm lens on the sample surface was about 120 μm (FWHM). An interference filter for 675–680 nm was placed at the entrance of a collimator Thorlabs F220FC-1064, which collected the laser diode radiation passed through the sample into the measuring optical fiber. To minimize the influence of spontaneous luminescence, the collecting collimator was located at a distance of about 200 mm from the sample. The spontaneous luminescence level, which enters the collimator without switching on the probe diode, was no more than 5% of the probe signal level and was subtracted when processing the stimulated emission gain. The stimulated emission was detected by a LeCroy WaveMaster 808Zi-A oscilloscope with a LeCroy OE555 optoelectronic converter with 4.5 GHz sample rate, which allows to study in real time the dynamics probe amplification in the sample under picosecond laser pump.

In the second series of experiments a cylindrical lens was employed to create an elliptical pump beam profile illuminating the whole zone S2 with surface area of about 4 × 0.25 mm. Since the pump beam area was increased significantly, we apply the full train of 15 picosecond pulses, instead of a single pump pulse. The sample was placed at the Brewster's angle to the probe beam, so that the probe beam was going along the diamond parallel to sample side surfaces (Fig. 4, inset 2). Note, that conventional Q-switched nanosecond Nd:YAG lasers usually have a stochastic structure of resonator light field time profile that leads to

significant pulse-to-pulse intensity variations with cavity round-trip period features. Such pump pulses cannot allow a controlled uniform pumping of the sample. In opposite, the train of active mode-locked picosecond laser pulses employed in our experiments has high temporal regularity and shot-to-shot repeatability[39].

The kinetics of the probe laser diode signal transmitted through the sample zones S1 and S2 are shown in Fig. 5a and b, respectively. The signal was normalized to the level of the unperturbed laser diode signal transmitted through the sample in the absence of a pump pulse. The 532-nm picosecond pump pulse arrives at moment in time $t = 0$, and leads to a disturbance of the detected 675-nm signal.

In the S1 zone, characterized by the presence of the NV$^0$ centers, the appearance of a pump pulse leads to an attenuation of the probe signal at all the considered pump energies (Fig. 5a). The leading edge of the transient response has a duration of less than 100 ps, which corresponds to the profile of the pump pulse. The relaxation process of the pump-induced absorption is essentially nonlinear and its characteristic time depends on the energy of the pump pulse. For pump energy below 1 μJ, the relaxation time is about 20 ns; for higher pump energies, the complete relaxation was not reached at times up to 10 μs. In this case, the relaxation process cannot be decomposed into fast and slow components and looks rather like a multi-particle process with a smooth change in the relaxation rate.

In the S2 zone, characterized by the predominance of the NV$^-$ centers, we registered the pump induced amplification of the probe signal (Fig. 5b). For pump energies below 7 μJ, the relaxation process of the pump-induced amplification has a characteristic time about 10 ns (Fig. 5b, black, blue and green curves), which is close to the lifetime of the NV$^-$ excited state, since the probe beam has a relatively weak intensity. The maximum registered gain is about 1.05 at a pump pulse energy of 6.6 μJ. Taking the sample depth as 0.25 mm and stimulated emission cross-section $\sigma_{em} = 4.9 \cdot 10^{-17}$ cm$^2$ (see the first section), the concentration of the pumped NV$^-$ centers reaches $0.4 \cdot 10^{17}$ cm$^{-3}$ that is about 16% from the total concentration of the NV$^-$ centers in the S2 zone.

At a pump energy of 15 μJ, the ionization process of the NV$^-$ centers and corresponding growth of the NV$^0$ concentration leads to a transition from the probe signal amplification to the probe signal absorption that occurs on a 1.5-ns time scale (Fig. 5b, red curve). Signal absorption at this pump energy indicates the formation of free carriers upon ionization of NV$^-$ and NV$^0$ centers in the S2 zone (see Fig. 3b). The dynamics of the induced-

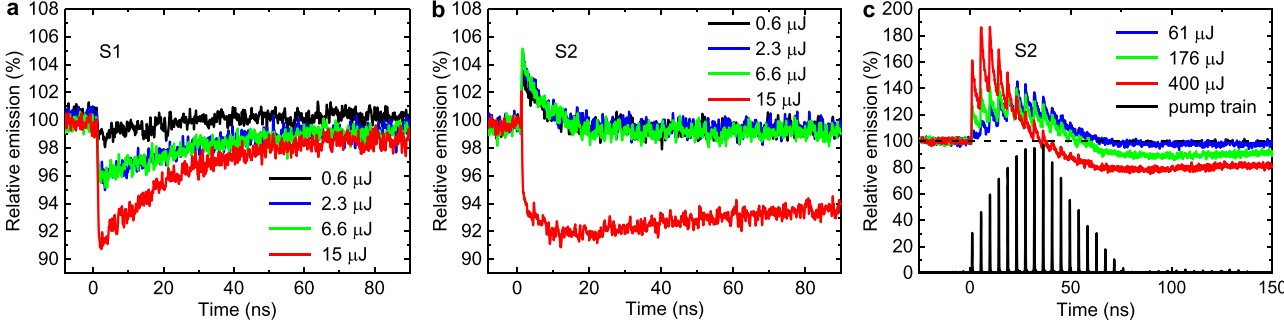

**Fig. 5 Stimulated emission from diamond sample.** Kinetics of the laser diode signal in a wavelength range from 675 nm to 680 nm transmitted through the sample zones S1 (**a**) and S2 (**b**) pumped by a single 150-ps 532-nm pulse at various energies: 0.6 µJ (black curve), 2.3 µJ (blue curve), 6.6 µJ (green curve), 15 µJ (red curve) in case of transverse probe configuration (Fig. 4, inset 1). **c** Kinetics of the 675-nm laser diode signal transmitted through the sample zone S2 pumped by a train of 150-ps 532-nm laser pulses at various energies: 61 µJ (blue curve), 176 µJ (green curve), and 400 µJ (red curve) in case of longitudinal probe configuration (Fig. 4, inset 2). Corresponding train of pump pulses with normalized amplitudes is plotted as a black curve. The front weak pulses of the train were dumped by a pulse selector.

absorption relaxation process is identical to the dynamics in the S1 zone. Thus, the process of induced absorption correlates with the presence of NV$^0$ centers in the studied sample, regardless of whether they were present initially or were formed as a result of pulse-induced ionization of NV$^-$ centers.

From the obtained kinetics of the probe signal, it is clear that there is a pump rate threshold, an exceeding of which leads to saturation of probe gain and further switch to an attenuation of the probe signal. Due to the demonstrated pump-saturation behavior of the probe gain, we consider a side pump configuration along the sample allowing us to reach a higher gain level. A cylindrical lens was employed to create an elliptical pump beam profile illuminating the crystal zone S2 with surface area of about 4 × 0.25 mm. Since the pump beam area was increased significantly, we apply the full train of 15 picosecond pulses, instead of a single pump pulse. The sample was placed at the Brewster's angle to the probe beam, so that the probe beam was going along the diamond parallel to sample side surfaces (Fig. 4, inset 2). The S1 zone of the sample was not considered in this series of experiments, due to no stimulated emission was observed in this zone.

Recorded kinetics of the probe signal transmitted through the sample shown in Fig. 5c for various energies of a multi-pulse pump. At low pump energy of 61 µJ, the probe amplification profile repeats the intensity profile of the pump pulse train (Fig. 5c, blue curve). At higher pump energy of 176 µJ, weak probe absorption appears at the end of the pump train (Fig. 5c, green curve). A saturation of probe amplification followed by NV$^-$-ionization induced absorption occurs at sufficiently high pump energy of 400 µJ (Fig. 5c red curve). Further increase of the pump energy will lead to an over-threshold pumping regime that does not provide optimal lasing conditions. The maximum achieved gain is 1.8 (gain coefficient 1.5 cm$^{-1}$), which is quite sufficient for stable lasing conditions. The corresponding concentration of exited NV$^-$ centers reaches $0.3 \cdot 10^{17}$ cm$^{-3}$, that is about 12% from the total concentration of the NV$^-$ centers in the S2 zone.

**Lasing on NV$^-$ centers in diamond**. Herein we report the scheme of the resonator, used for demonstration of the lasing on NV$^-$ centers in diamond, and the characterization of the lasing pulses. The resonator consists of two identical flat mirrors with reflection coefficient 95% in the range of wavelengths from 700 to 750 nm. The diamond sample was aligned at the Brewster's angle in resonator that provides a lasing direction along the sample. In

order to compensate the beam divergence and to increase the resonator stability, a BK7 spherical lens with focal length of 15 mm was introduced in the resonator at the Brewster's angle to the beam axis (Fig. 6a). The cavity length was chosen to provide a symmetrical mode on one of the mirrors (Fig. 6d). We were operating with the sample zone S2, in which the stimulated emission has been obtained. A full train of 150-ps 532-nm laser pulses was used to pump the sample. As in the previous section, the cylindrical lens was employed for side illumination of the whole S2 zone of the sample.

A laser generation at NV$^-$ color centers in diamond was obtained under pump energies of ps-pulse train above 20 µJ. Dependence of output laser pulses energy (sum of energies from two couplers) on pump energy is presented in Fig. 7a. The maximal energy of the achieved laser pulses reaches 10.4 nJ (5.2 nJ at each output coupler) under 260-µJ pump. With a further increase of pump rate, the output pulses energy decreases due to NV centers ionization and formation of an induced plasma. Spatial profiles of lasing mode were recorded at 200-mm distances from each output coupler by the Ophir SP620U CCD camera and are shown in Fig. 6b and c. The polarization measurements show that the laser beam is linear polarized according to the Brewster's angle of diamond sample.

Temporal profiles of the lasing pulse at various pump energies are given in Fig. 7b. The minimal lasing pulse duration of about 1 ns (FWHM) has been achieved for pump energies from 73 µJ to 140 µJ (Fig. 7b violet and blue curves). The active media inversion population changes dramatically on the time scale of a delay between the neighboring pulses in the pump train because the lifetime of the NV$^-$ excited state (about 10 ns) is comparable to the 4.4-ns delay between pump pulses in the train. In this case the minimal laser pulse duration can be achieved if the laser generation develops in resonator after the arrival of the second pulse from the pump pulse train. At high pump energies (260 µJ and 400 µJ), the lasing development occurs between the first and the second pump pulses of the train and corresponding lasing pulse time profile consists of several peaks (Fig. 7b red and black curves).

Despite the fact that the fluorescence peak of NV$^-$ centers in diamond lies at a wavelength of 680 nm (Fig. 2c), the laser pulse spectrum has a maximum at 720 nm (Fig. 7c). This is not a contradiction, since the gain cross-section spectrum can be obtained by multiplying the fluorescence emission spectrum by a factor of $\lambda^5$ and, therefore, the gain maximum is shifted to the long-wavelength region[43]. Similarly, for the well-known Ti:Sapphire laser the fluorescence emission peak is at 750 nm, while the gain maximum is located near 800 nm[43].

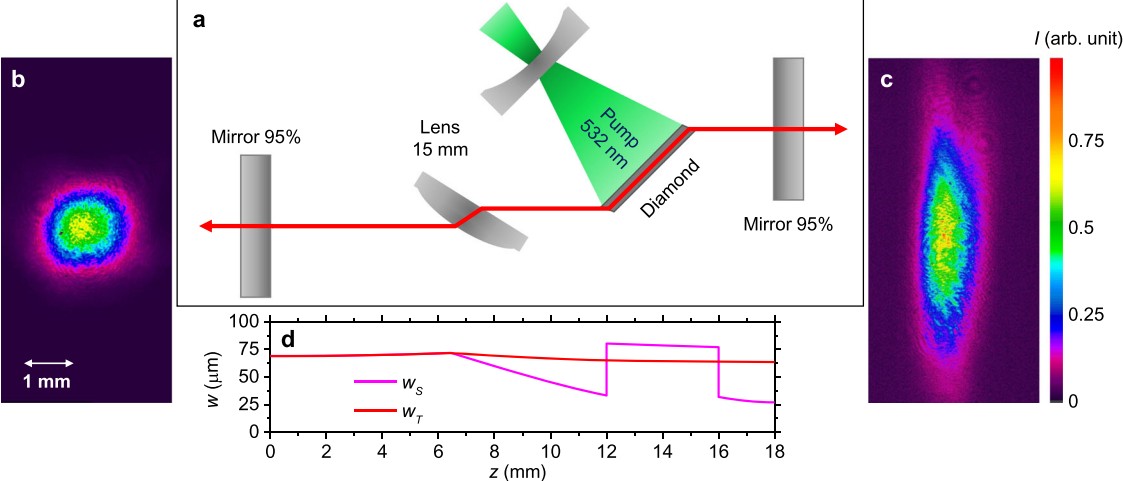

**Fig. 6 NV⁻ diamond laser. a** Resonator scheme of the NV⁻ color centers diamond laser. **b, c** Beam profiles at a distance of 200 mm from the resonator mirrors. **d** Calculated transverse mode radius $w_T$ and sagittal mode radius $w_S$ along the resonator (intensity e$^{-2}$ level).

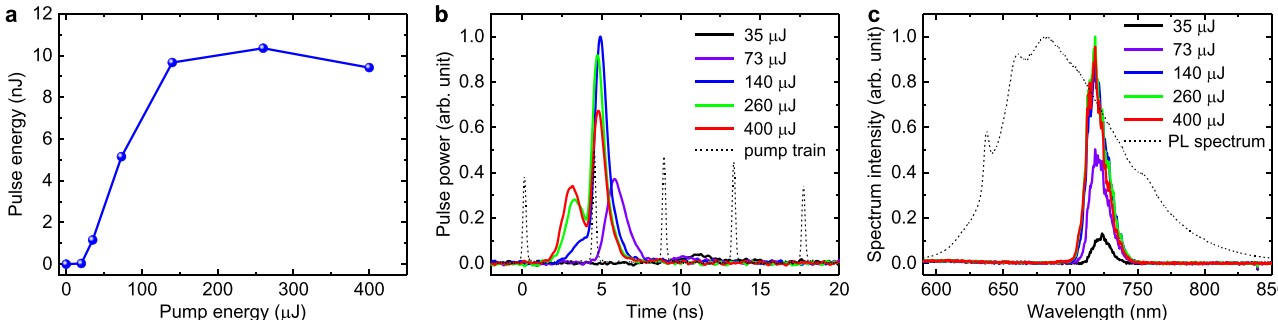

**Fig. 7 Lasing pulse characteristics.** Laser pulses energy (sum of energies from two couplers) versus the pump (**a**), time profiles of lasing pulses (**b**), and lasing spectra (**c**) at various pump pulse train energies: 35 μJ (black curve), 73 μJ (violet curve), 140 μJ (blue curve), 260 μJ (green curve) and 400 μJ (red curve). Black dashed curves: train of pump pulses with normalized amplitudes (**a**) and photoluminescence spectra of S2 zones of diamond sample under continuous excitation at 532 nm (**b**).

The laser pulse spectrum width is about 20 nm (FWHM). A lasing process starts from noise in the entire gain band and the pulse rise time in the cavity (~5 ns, 20–30 roundtrips) is sufficiently for lasing spectrum narrowing by about an order in comparison with luminescence spectrum width (Fig. 7c).

## Discussion

Comparative spectroscopic studies have proven the possibility of lasing in diamond samples containing NV⁻ color centers with densities about $10^{17}$ cm⁻³. The presence of NV⁰ centers, regardless of whether they were initially contained in the sample or were formed during the NV⁻ ionization process under a high-intensity laser pumping, leads to a decrease of the stimulated emission and a transition to the probe absorption regime. We conclude that for 532-nm pump wavelength there is an optimal pumping rate at which the highest gain in the NV⁻ diamond is achieved. The maximum reached gain is 1.8 (gain coefficient 1.5 cm⁻¹) when the sample zone containing NV⁻ centers was pumped by a train of picosecond pulses. Stable lasing at a wavelength of 720 nm was achieved in diamond sample under optimal pumping conditions. Compelling experimental evidence on lasing were observed: 10-nJ pulse energy, a near diffraction limited laser beam, linear polarization, nanosecond pulse temporal profile, and lasing spectrum narrowing by about an order in comparison with luminescence spectrum width.

In our vision the next important step in future investigations is CW lasing in diamond samples with NV⁻ color centers. The movement towards the quasi-CW and CW lasing should be carried out by gradually increasing the duration of the pumping pulse, while carefully controlling the gain and induced losses ratio. One of the problems reaching CW lasing is the absorption in the induced plasma during the ionization of the NV centers by pump radiation.

The achieved lasing at NV⁻ color centers in diamond will lead to an enhancement of a broad laser system development based on the NV⁻ diamond active media. A unique combination of properties such as thermal conductivity, low thermal expansion, and mechanical strength makes diamond crystals an excellent candidate for high-power CW and ultrashort lasers.

## Methods

**Diamond sample preparation.** A diamond sample is a $4 \times 3 \times 0.25$ mm plate cut from HPHT diamond subjected to a radiation treatment by electrons accelerated to 3 MeV with a dose of about $1 \cdot 10^{18}$ e⁻/cm² and subsequent annealing in vacuum at a temperature of 800 °C for 24 h. Two lateral sides of the sample were polished at 22.5-degrees angle to enable studies of the probe amplification incident on the sample at the Brewster's angle (67.5 degrees).

The diamond sample has two growth zones with significantly different concentrations of nitrogen, which is incorporated into the diamond lattice during HPHT synthesis. Most of the diamond sample has a nitrogen concentration of less than 1 ppm (zone S1), while the narrow colored area at the edge of the sample has a nitrogen concentration that is about 2 orders of magnitude higher (zone S2). NV complexes are formed in the crystal after irradiation with 3 MeV electrons and annealing in vacuum at 800 °C. In the S1 sample zone with a low nitrogen content, a significant fraction (about a half) of nitrogen atoms have formed impurity-defect complexes (NV centers) with vacancies in a neutral charge state. In the zone of the diamond sample with an increased concentration of nitrogen S2 (colored region),

the main part of nitrogen remained in the form of single substitutional atoms, which are a donor impurity. In S2 zone, excess electrons from some of the substitutional nitrogen atoms were captured by NV centers. Thus, in the colored region of sample S2, NV centers in the negative charge state were formed, while keeping a negligible fraction of original NV centers in the neutral charge state.

**Emission and absorption cross-sections of the sample**. For S2 zone of the sample, we estimate emission and absorption cross-sections based on the spectroscopic sample characteristics (Fig. 2). Here we take the fluorescence quantum efficiency being close to 1. The emission cross section was calculated at the center of the emission band $\lambda_{em} = 680$ nm assuming a Gaussian emission line shape in the frequency domain[44–48]:

$$\sigma_{em} = \frac{\lambda_{em}}{4\pi n^2 \tau_{21} \Delta \nu_{em}} \left( \frac{\ln 2}{\pi} \right)^{\frac{1}{2}}, \tag{1}$$

where emission band is fitted by a Gaussian line shape with a width FWHM $\Delta\nu = 66$ THz that corresponds to $\Delta\omega = 0.27$ eV ($2.2 \cdot 10^3$ cm$^{-1}$) and $\omega_0 = 1.81$ eV ($1.46 \cdot 10^4$ cm$^{-1}$). The estimated emission cross-section value reaches $\sigma_{em} = 4.9 \cdot 10^{-17}$ cm$^2$. We estimate absorption cross section at the center of the absorption band $\lambda_{abs} = 569$ nm assuming also a Gaussian emission line shape in the frequency domain[44–48]:

$$\sigma_{abs} = \frac{g_2}{g_1} \frac{\lambda_{abs}}{4\pi n^2 \tau_{21} \Delta \nu_{abs}} \left( \frac{\ln 2}{\pi} \right)^{\frac{1}{2}}, \tag{2}$$

where $g_1 = 1$ and $g_2 = 2$ are degeneracies of ground and excited states, respectively, absorption band is fitted by a Gaussian line shape with a width FWHM $\Delta\nu = 78$ THz, that corresponds to $\Delta\omega = 0.33$ eV ($2.7 \cdot 10^3$ cm$^{-1}$) and $\omega_0 = 2.16$ eV ($1.7 \cdot 10^4$ cm$^{-1}$). The estimated absorption cross-section value reaches $\sigma_{abs} = 5.7 \cdot 10^{-17}$ cm$^2$.

## Data availability

The processed numerical data files are available at the public GitHub repository (https://github.com/VMitrokhin/Diamond-Lasing). All other data that support the plots within this paper and other findings of this study are available from the corresponding author upon reasonable request.

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

## Acknowledgements

EL and DG acknowledge the Ministry of Education and Science of Russian Federation for funding the research (the state order, project No 0721-2020-0048).

## Author contributions

A.S., A.D. and E.L. conceived and planned the experiments. A.Y., E.L. and D.G. carried out the spectroscopic characterization of the diamond sample. A.S., A.D., V.M. and S.P. carried out the main experiments. A.Y. and V.V. contributed to diamond sample preparation. A.D., E.L., E.S. and A.S. contributed to the interpretation of the results. A.D. and E.S. took the lead in writing the manuscript. All authors provided critical feedback and helped shape the research, analysis and manuscript.

## Competing interests

The authors declare no competing interests.
