## [Peer review file · Nature Communications]

REVIEWER COMMENTS

Reviewer #1 (Remarks to the Author):

Manuscript title: NV- Diamond Laser

The authors claim to have demonstrated NV- diamond laser for the first time. Authors have also referred to earlier works in references 2 and 27. Indeed, reference #27 (Indirect measurements without optical gain and lasing action) and a few other references provide the possibility of NV-diamond laser. Hence, the work is incremental to reference #27. In addition, reference #27 was published in Nature Communications with a possibility of lasing action.

However, the experimental demonstration is something that is novel in the manuscript and if authors wish, the manuscript may be transferred to Scientific Reports with the following comments, being addressed.

1) Figure 1 should be in supplemental information as it does not provide key information; however, if the reader wants to know may refer to the supplemental information.

2) The Y-axis of the absorption spectrum in Fig. 2(a) should be in OD (Optical Density). If authors quote in the units of cm^{-1} , they need to provide complete details regarding the conversion. Also, the authors need to increase the figure size. Further, the details on photoluminescence spectra shown in Fig 2(b) and 2(c) need to be provided in supplemental. Were these spectra acquired using conventional fluorimeter and if so, the details of the company and spectral resolution, etc.

3) Authors reported lasing action at 10^{17} cm^{-3} (NV- centers density). It is under critical plasma density. How did the authors get this number? What if this number goes above critical plasma density? Is there any role of plasma lasing?

Referee

Reviewer #2 (Remarks to the Author):

The present manuscript by Savvin et al reports on the observation of pulsed laser action from negatively charged nitrogen-vacancy centres (NV-) in diamond. As such, the result is an important milestone since lasing action from NV- centres in diamond has been an open goal in the field of diamond quantum/laser science for at least a couple of decades. I congratulate the authors on their achievement and would in principle support publication of the results in Nature Communications. However, there are a few important changes to be made before the manuscript can be published:

The authors of the present study claim the first direct demonstration of stimulated emission of radiation from NV- centres in diamond. While the data are sound, the claim that stimulated emission of radiation has not been demonstrated before is not correct. There are two references in literature that demonstrate stimulated emission of radiation from NV- centres in diamond, only one of which the authors cite: Ref[27] in the present manuscript (Jeske et al., Nature Comm) uses a lock-in technique to demonstrate stimulated emission of radiation. The authors of the present study argue that this is an indirect way of detecting stimulated emission. I would object to this statement but admit that this is somewhat subjective and a matter of 'taste'.

However, a study that appeared in September 2020 in Nanophotonics (vol. 9, no. 15, 2020, pp. 4505-4518) directly and unambiguously demonstrated the amplification of radiation by stimulated emission

from NV⁻ centres in diamond inside an optical cavity. The method of amplification of a weak probe beam in the presence of a green 532nm laser is the same method as used in the present manuscript. Though the Nanophotonics paper does not show standalone lasing, realisation of optical gain with CW measurements is an important step for realising a CW NV⁻ laser. I therefore strongly recommend the authors to reference that work and rectify their claim.

The one key result of the present work is the observation of stimulated emission of spontaneous radiation, or spontaneous lasing, under pulsed pumping. This claim is well justified and the data presented by the authors are sound. However, the article falls significantly short of motivating the importance of the findings and providing sufficient context for the reader to understand the implications. In my point of view, the significance of realizing an NV⁻ laser goes far beyond being simply a new addition to the existing 'zoo' of lasers.

The achievement of any type of lasing from NV⁻ centres in diamond is a long-awaited result and paves the way for interesting (quantum) applications: an NV⁻ laser has the potential to be used as a room-temperature magnetometer which can have a sensitivity of a few $\text{fT}/\sqrt{\text{Hz}}$ (New Journal of Physics 18.1 (2016): 013015.). Though such a magnetometry scheme ultimately requires a continuous-wave (CW) NV⁻ laser, an achievement of a pulsed NV⁻ laser is a key breakthrough for realising such high-sensitivity magnetometers since it allows for a systematic exploration of parameters required for NV⁻ lasing under realistic experimental conditions.

Given the NV⁻ laser reported in the present work is operating in pulsed mode only, the search for realising a CW NV⁻ laser is not over yet. Hence it would be better to clearly indicate, even in the title, that the article reports on a pulsed NV⁻ laser. Could the authors also clarify their motivation behind trying for a pulsed laser rather than a CW laser?

Finally, a few more technical questions:

It is interesting to see that in Ref [Nanophotonics, vol. 9, no. 15, 2020, pp. 4505-4518], the optical gain is seen around 721nm (which is set by the external stimulating wavelength), which is very close to the 720nm central lasing wavelength reported in the present article. Could the authors give an interpretation as to why the lasing wavelength is 720 nm, though they have 95% reflection for cavity mirrors in the range 700nm - 750nm?.

In page 4 the authors mention "We observe strong blue absorption in the spectral range from 400 to 495 nm (Fig. 2a, black curve) that indicates a significant amount of nitrogen atoms in the S2 zone (Reference 31)." It is not clear how they have arrived at this conclusion. It seems like the effect shown in Figures 3b and 3c may also be interpreted in light of single-photon induced ionisation from NV⁻ to NV⁰ and the recombination process. Could the authors please provide some clarity on this aspect?

The S2 which the authors chose for investigating lasing seems to have low photo-ionisation, and maybe that is the key factor. Could the authors provide a bit of clarity on the stimulated emission rate in the laser cavity versus the typical ionisation rate (which might be low in their case)?

In order to substantiate the claim of lasing, usually a threshold curve is presented (that is output power versus pump power). Some of that information is contained in Figure 7, but is not very obvious. I encourage the authors to present a corresponding threshold curve.

The manuscript needs to be carefully proof-read. Apart from the occasional minor typo, there are some inconsistencies such as for example the quoted maximal gain coefficient (in the introduction it is 1.4 cm^{-1} , later in the text the value is given as 1.5 cm^{-1}).

Reviewer #3 (Remarks to the Author):

Despite the proposal of a laser based on diamond defects is dating back to several decades ago, it has never been experimentally demonstrated. The paper by Savvin and coworkers presents the first direct measurements of the optical gain and lasing of the negatively charged nitrogen-vacancy (NV) centers in diamond. The progress can be, among others, attributed to usage of a high-power custom-made pulsed Nd:YAG laser with active phase stabilization. The presented data is mostly of high quality and convincing (with the exceptions listed below), and the paper is generally well written and has a good structure. Before the main results are discussed and shown there is a thorough sample characterization that helps to support the main conclusions and potentially reproduce the experiment in other labs.

However, some points need to be addressed before the paper could be published. The points are listed below. I have also updated the "Laser reporting summary" (attached) with my comments.

1. The key information missing (or not complete) to fully support the claim of lasing regards the output power vs. input power plot, which is only present indirectly (spectra for different pump power are shown). Only one spectrum is shown around the threshold (at 35 μJ). The lasing seems to saturate quickly and there are only two measurements at or above the threshold and below saturation (35 and 73 μJ). More experimental points in this area would help to make the experimental data more convincing. Also, the spectrum below the threshold is normalized, which excludes any quantitative comparisons. More data points and the plot of integrated spectra vs. pump power would be of great value here.

2. Authors write: "... peak intensity of single pulse is $3 \cdot 10^9 \text{ W/cm}^2$...". Therefore, this pumping mode delivers a laser-matter interaction regime that drastically differs from the low-power CW laser pump discussed in the previous section. This statement is vague to me. Please specify, in which physical way this interaction regime is different and which role here it plays? In my understanding, the effect of the laser illumination in this work is limited to exciting the NV center electronic configurations to their excited states and exciting electrons from the valence to intraband defect levels (NV-related and probably others) and further to the conduction band. In my understanding, the only difference between the CW and pulsed laser is the rate at which this excitation (and thus population inversion) occurs, which indeed is a new regime that allows lasing. I think the wording "interaction regime" might be misleading here especially for communities studying other phenomena related to light-matter interactions (such as those in atomic physics or laser machining).

3. Authors write: "At a pump energy of 15 μJ , the ionization process of the NV⁻ centers and corresponding growth of the NV0 concentration leads to a transition from the probe signal amplification to the probe signal absorption that occurs on a 1.5-ns time scale (Fig. 5b, red curve)". I think this requires more explanation. What is the origin of the absorption? 675 nm is typically outside of both NV⁻ and NV0 absorption range.

4. Authors write: "To minimize the influence of spontaneous luminescence, the collecting collimator is located at a distance of about 200 mm from the sample". I am not convinced that this suffice to minimize the influence of spontaneous luminescence to a negligible level. How big would be the detected luminescence from the green pump as compared to the detected stimulated emission? This could be easily checked experimentally and would help to eliminate any alternative explanations of the observed effects.

5. The lasing occurs around 720 nm, which seems to be the wavelength giving the highest gain. Why do the authors test the stimulated emission around 675 nm then? Would 720 nm give better results (higher gain)? What was the motivation for choosing 675 nm?

Only the first of the deficiencies mentioned above requires a major change to the manuscript. I consider the other ones as minor deficiencies. Once the raised issues are addressed (either by new experiments or thorough explanations), I am convinced that this manuscript will fully merit publication in Nature Communications.

Further suggestions to improve the manuscript.

Below, I have listed other comments that may help to improve the manuscripts (please use them as mere suggestions, no need to address them) and help to understand the underlying physics.

I. The potential significance of the result is high. However, more motivation and perspective would generally help the general reader to understand this significance better.

II. Describe to the general reader what inhibited the development of defect-based lasers in diamond so far. The idea is very old, but there is no information on what were the major obstacles. How important is the use of the custom-made pulsed Nd:YAG laser? Are there any comparable commercial devices? The link between active mode-locking, possibility of controlled uniform (temporal?) pumping of the sample and observation of lasing is unclear to me.

III. Fit the data on Fig. 5 with exponential functions and plot the amplitude and decay constant vs. pulse energy – I think that this could help to extract and visualize the essential physical information from these figures. I believe this might be nontrivial and in case of panel (b) and (c) and rate-equations might be necessary to model the data.

IV. Energetically, the lasing (around 720 nm, that is 1.72 meV) and pumping (2.33 meV) occur at a roughly comparable energetic distance from the zero phonon line (1.95 meV). This means that the emission and absorption are assisted by the excitation of a vibrons of comparable energies. Do authors think that this could be anyhow relevant?

V. There are a couple of typos, like:

which many orders above the CW lasers => which is many orders above the CW lasers
Nothing major or excessive, a careful check and using spell-checker should help.

Dear Editors!

We are grateful to Reviewers for reading of our manuscript and helpful remarks. Just below there is letter responding to the reviewers of our Manuscript #: **NCOMMS-21-08760** to Nature Communications entitled “**NV⁻ Diamond Laser**” and noting the changes that have been made in the text according to Reviewers comments.

Reviewer #1:

1) Figure 1 should be in supplemental information as it does not provide key information; however, if the reader wants to know may refer to the supplemental information.

We believe that the Figure 1 is necessary in the main text of the manuscript, since it shows a typical photograph of the treated HPHT diamond used in experiments and illustrates the presence in our sample of two Zones with different concentrations of NV⁻ centers.

2a) The Y-axis of the absorption spectrum in Fig. 2(a) should be in OD (Optical Density). If authors quote in the units of cm⁻¹, they need to provide complete details regarding the conversion. Also, the authors need to increase the figure size.

It is done in our corrected version:

“The absorption spectra after Fresnel correction are given in Fig. 2a in the units of cm⁻¹ (gray curve for S1 and black curve for S2 sample zones, respectively). The absorption coefficient $\alpha(\lambda) = \frac{1}{d} \ln \frac{I_0(\lambda)}{I(\lambda)}$ was obtained by illuminating the sample by a tungsten incandescent lamp with incident intensity $I_0(\lambda)$ and transmitted intensity $I(\lambda)$ through the sample with thickness $d = 0.25$ mm for each zone.” (page 4, first paragraph)

2b) Further, the details on photoluminescence spectra shown in Fig 2(b) and 2(c) need to be provided in supplemental. Were these spectra acquired using conventional fluorimeter and if so, the details of the company and spectral resolution, etc.

It is done in our corrected version:

“Transmitted intensity $I(\lambda)$ was measured at room temperature employing an Ocean Optics FLAME-S-XR1 spectrometer with optical resolution 1.69 nm FWHM and 200- μ m fiber input for light collection.” (page 4, first paragraph)

“Normalized photoluminescence spectra of S1 and S2 zones recorded at room temperature employing the Ocean Optics FLAME-S-XR1 spectrometer are given in Fig. 2b and c, respectively.” (middle of page 5)

3) Authors reported lasing action at 10¹⁷ cm⁻³ (NV centers density). It is under critical plasma density. How did the authors get this number? What if this number goes above critical plasma density? Is there any role of plasma lasing?

The concentration N_{NV^-} of NV⁻ centers was obtained using the absorption cross-section σ_{abs} and corresponding absorption coefficient from Fig. 2a:

“Using the obtained absorption cross-section σ_{abs} and corresponding absorption coefficient α from Fig. 2a, the concentration $N_{NV^-} = \frac{\alpha}{\sigma_{abs}d}$ of NV^- centers is found to be $\sim 2.5 \cdot 10^{17} \text{ cm}^{-3}$ (1.4 ppm) in the S2 zone.” (bottom of page 6).

We believe that NV centers form enough deep levels in diamond and are not a source of donor electrons (plasma) at room temperature. However, under intense pumping, the induced absorption recorded by us indicates the formation of free carriers upon ionization of NV centers, which prevents lasing.

Reviewer #2:

a) The authors of the present study claim the first direct demonstration of stimulated emission of radiation from NV- centres in diamond. While the data are sound, the claim that stimulated emission of radiation has not been demonstrated before is not correct. There are two references in literature that demonstrate stimulated emission of radiation from NV- centres in diamond, only one of which the authors cite: Ref[27] in the present manuscript (Jeske et al., Nature Comm) uses a lock-in technique to demonstrate stimulated emission of radiation. The authors of the present study argue that this is an indirect way of detecting stimulated emission. I would object to this statement but admit that this is somewhat subjective and a matter of ‘taste’.

However, a study that appeared in September 2020 in Nanophotonics (vol. 9, no. 15, 2020, pp. 4505-4518) directly and unambiguously demonstrated the amplification of radiation by stimulated emission from NV- centres in diamond inside an optical cavity. The method of amplification of a weak probe beam in the presence of a green 532nm laser is the same method as used in the present manuscript. Though the Nanophotonics paper does not show standalone lasing, realisation of optical gain with CW measurements is an important step for realising a CW NV- laser. I therefore strongly recommend the authors to reference that work and rectify their claim.

We agree with the Reviewer and add in our corrected version a description of the experiment on stimulated amplification in a sample placed in a fiber microcavity and abandon the wording “indirect way of detecting stimulated emission”:

“A rise of stimulated emission at wavelengths in the PSB of NV^- zero-phonon line (ZPL) was detected by measuring a decrease of the NV^- spontaneous emission under a CW optical pump at 532 nm. In the recent work³² the amplification of 721-nm laser diode radiation by stimulated emission from NV^- centres in diamond inside an optical fiber cavity was demonstrated. However a lasing at NV^- centers in diamond have not been delivered yet.” (bottom of page 2)

b) The one key result of the present work is the observation of stimulated emission of spontaneous radiation, or spontaneous lasing, under pulsed pumping. This claim is well justified and the data presented by the authors are sound. However, the article falls significantly short of motivating the importance of the findings and providing sufficient context for the reader to understand the implications. In my point of view, the significance of realizing an NV- laser goes far beyond being simply a new addition to the existing ‘zoo’ of lasers.

The achievement of any type of lasing from NV⁻ centres in diamond is a long-awaited result and paves the way for interesting (quantum) applications: an NV⁻ laser has the potential to be used as a room-temperature magnetometer which can have a sensitivity of a few fT/ $\sqrt{\text{Hz}}$ (New Journal of Physics 18.1 (2016): 013015.). Though such a magnetometry scheme ultimately requires a continuous-wave (CW) NV⁻ laser, an achievement of a pulsed NV⁻ laser is a key breakthrough for realising such high-sensitivity magnetometers since it allows for a systematic exploration of parameters required for NV⁻ lasing under realistic experimental conditions.

We hope that the results of our research presented in our manuscript will allow in the near future to create not only new CW and high-power ultrashort lasers (page 15), but also paves the way for important quantum applications. We have added in the introduction a reference to the work (New Journal of Physics 18.1 (2016): 013015.):

“Laser initialization, changing and reading the spin state of an NV⁻ electron allows to control the state of a quantum system and turn NV⁻ centers in diamond into a promising platform for optical quantum computing¹²⁻¹⁹, quantum metrology, magnetometry, sensing and visualization²⁰⁻²⁶.” (page 2, ref. 20).

c) Given the NV⁻ laser reported in the present work is a operating in pulsed mode only, the search for realising a CW NV⁻ laser is not over yet. Hence, it would be better to clearly indicate, even in the title, that the article reports on a pulsed NV⁻ laser. Could the authors also clarify their motivation behind trying for a pulsed laser rather than a CW laser?

To answer this remark, we add a new paragraph in the manuscript text:

“In our vision the next important step in future investigations is CW lasing in diamond samples with NV⁻ color centers. The movement towards the quasi-CW and CW lasing should be carried out by gradually increasing the duration of the pumping pulse, while carefully controlling the gain and induced losses ratio. One of the problems reaching CW lasing is the absorption in the induced plasma during the ionization of the NV centers by pump radiation.” (bottom of page 15)

Since lasing in diamond samples containing NV⁻ color centers was registered in the first time, we did not specify in the manuscript title that the pumping and generated laser pulses have picosecond and nanosecond durations, respectively.

Finally, a few more technical questions:

1) It is interesting to see that in Ref [Nanophotonics, vol. 9, no. 15, 2020, pp. 4505-4518], the optical gain is seen around 721nm (which is set by the external stimulating wavelength), which is very close to the 720nm central lasing wavelength reported in the present article. Could the authors give an interpretation as to why the lasing wavelength is 720 nm, though they have 95% reflection for cavity mirrors in the range 700nm - 750nm?

To answer this remark, we add the paragraph in the manuscript text:

“Despite the fact that the fluorescence peak of NV⁻ centers in diamond lies at a wavelength of 680 nm (Fig. 2c), the laser pulse spectrum has a maximum at 720 nm (Fig. 7c). This is not a contradiction, since the gain cross-section spectrum can be obtained by multiplying the fluorescence emission spectrum by a factor of λ^5 and, therefore, the gain maximum is shifted to

the long-wavelength region⁴³. Similarly, for the well-known Ti:Sapphire laser the fluorescence emission peak is at 750 nm, while the gain maximum is located near 800 nm⁴³." (middle of page 14)

2) In page 4 the authors mention "We observe strong blue absorption in the spectral range from 400 to 495 nm (Fig. 2a, black curve) that indicates a significant amount of nitrogen atoms in the S2 zone (Reference 31)." It is not clear how they have arrived at this conclusion. It seems like the effect shown in Figures 3b and 3c may also be interpreted in light of single-photon induced ionisation from NV⁻ to NV⁰ and the recombination process. Could the authors please provide some clarity on this aspect?

We have added to the text of the manuscript information on the concentration of nitrogen in S2:

"We observed strong blue absorption in the spectral range from 400 to 495 nm (Fig. 2a, black curve) that indicated a significant amount of nitrogen atoms in the S2 zone³³. The nitrogen concentration $3.1 \times 10^{19} \text{ cm}^{-3}$ (175 ppm) in S2 zone was measured employing Bruker Vertex 70 FTIR spectrometer combined with a micro-scope Hyperion 2000." (bottom of page 4).

We and other researchers interpreted Figures 3b and 3c as photoionization from NV⁻ to NV⁰ and the recombination process. We indicated this result in the text of our manuscript:

"As a result, we observe a saturation of the NV⁻ contribution and appearance of the NV⁰ ZPL and PSB in the range of 575 – 630 nm in the photoluminescence spectrum (Fig. 3b, blue and red curves). The observed pump-saturation effect of NV⁻ photoluminescence is in a good agreement with the works^{27, 40-42}, where the role of the NV⁻ photo-ionization process under pump power increase was investigated." (bottom of page 7).

3) The S2 which the authors chose for investigating lasing seems to have low photo-ionisation, and maybe that is the key factor. Could the authors provide a bit of clarity on the stimulated emission rate in the laser cavity versus the typical ionisation rate (which might be low in their case)?

In the S2 zone, where the NV⁰ concentration is not high, the ionization rate is lower than in the S1 zone, where only induced absorption was observed (fig 5a). We believe that the probability of photoionization increases with an increase in the concentration of NV⁰ centers (fig 5b). We noted this statement in the text of our manuscript:

"Thus, the process of induced absorption correlates with the presence of NV⁰ centers in the studied sample, regardless of whether they were present initially or were formed as a result of pulse-induced ionization of NV⁻ centers. (middle of page 11)."

We also added in corrected manuscript the sentence about the correspondence of the pump energy of results from Fig. 3b and Fig. 5b:

"At a pump energy of 15 μJ , the ionization process of the NV⁻ centers and corresponding growth of the NV⁰ concentration leads to a transition from the probe signal amplification to the probe signal absorption that occurs on a 1.5-ns time scale (Fig. 5b, red curve). Signal absorption at this pump energy indicates the formation of free carriers upon ionization of NV⁻ and NV⁰ centers in the S2 zone (see Fig. 3b)." (middle of page 11)

4) In order to substantiate the claim of lasing, usually a threshold curve is presented (that is output power versus pump power). Some of that information is contained in Figure 7, but is not very obvious. I encourage the authors to present a corresponding threshold curve.

We added new Figure 7a with corresponding threshold curve (page 15):

Fig. 7a — Laser pulses energy (sum of energies from two couplers) versus the pump pulses energy

5) The manuscript needs to be carefully proof-read. Apart from the occasional minor typo, there are some inconsistencies such as for example the quoted maximal gain coefficient (in the introduction it is 1.4 cm^{-1} , later in the text the value is given as 1.5 cm^{-1}).

In the new version of the manuscript, we tried to correct all typos and inaccuracies.

Reviewer #3 (Remarks to the Author):

1) The key information missing (or not complete) to fully support the claim of lasing regards the output power vs. input power plot, which is only present indirectly (spectra for different pump power are shown). Only one spectrum is shown around the threshold (at $35 \mu\text{J}$). The lasing seems to saturate quickly and there are only two measurements at or above the threshold and below saturation (35 and $73 \mu\text{J}$). More experimental points in this area would help to make the experimental data more convincing. Also, the spectrum below the threshold is normalized, which excludes any quantitative comparisons. More data points and the plot of integrated spectra vs. pump power would be of great value here.

In the revised version of manuscript we added Figure 7a with corresponding threshold curve (Laser pulses energy versus the pump pulses energy). (Page 15)

2) Authors write: "... peak intensity of single pulse is $3 \cdot 10^9 \text{ W/cm}^2$...". Therefore, this pumping mode delivers a laser-matter interaction regime that drastically differs from the low-power CW laser pump discussed in the previous section. This statement is vague to me. Please specify, in which physical way this interaction regime is different and which role here it plays? In my understanding, the effect of the laser illumination in this work is limited to exciting the NV center electronic configurations to their excited states and exciting electrons from the valence to intraband defect levels (NV-related and probably others) and further to the conduction band. In my understanding, the only difference between the CW and pulsed laser is the rate at which this excitation (and thus population inversion) occurs, which indeed

is a new regime that allows lasing. I think the wording “interaction regime” might be misleading here especially for communities studying other phenomena related to light-matter interactions (such as those in atomic physics or laser machining).

We certainly agree with the reviewer. Although the mode of pulsed excitation of the medium in many cases differs from CW pumping, the process of saturation of luminescence and ionization of NV⁻ centers is observed even in CW pumping regime.

Therefore, we refused to use the term “interaction regime” in the text of our manuscript and removed the phrase:

~~“Therefore, this pumping mode delivers a laser-matter interaction regime that drastically differs from the low-power CW laser pump discussed in the previous section.” (top of page 9)~~

3) Authors write: “At a pump energy of 15 μJ, the ionization process of the NV⁻ centers and corresponding growth of the NV₀ concentration leads to a transition from the probe signal amplification to the probe signal absorption that occurs on a 1.5-ns time scale (Fig. 5b, red curve)”. I think this requires more explanation. What is the origin of the absorption? 675 nm is typically outside of both NV⁻ and NV₀ absorption range.

It is done in our corrected version:

“Signal absorption at this pump energy indicates the formation of free carriers upon ionization of NV⁻ and NV₀ centers in the S2 zone (see Fig. 3b).” (middle of page 11)

4) Authors write: “To minimize the influence of spontaneous luminescence, the collecting collimator is located at a distance of about 200 mm from the sample”. I am not convinced that this suffices to minimize the influence of spontaneous luminescence to a negligible level. How big would be the detected luminescence from the green pump as compared to the detected stimulated emission? This could be easily checked experimentally and would help to eliminate any alternative explanations of the observed effects.

It is done in our corrected version:

“The spontaneous luminescence level, which enters the collimator without switching on the probe diode, was no more than 5% of the probe signal level and was subtracted when processing the stimulated emission gain.” (top of page 9)

5) The lasing occurs around 720 nm, which seems to be the wavelength giving the highest gain. Why do the authors test the stimulated emission around 675 nm then? Would 720 nm give better results (higher gain)? What was the motivation for choosing 675 nm?

We used a 675-nm Thorlabs HL6750MG CW laser diode with a wavelength corresponding to the maximum of the sample S2 zone luminescence spectrum. The difference in the diode wavelength from 720 nm, at which lasing was observed, would not significantly affect the results of studies on amplification of the probe signal.

We also add in the manuscript the explanation of the shift between fluorescence peak (680 nm) and gain maximum (720 nm):

“Despite the fact that the fluorescence peak of NV⁻ centers in diamond lies at a wavelength of 680 nm (Fig. 2c), the laser pulse spectrum has a maximum at 720 nm (Fig. 7c). This is not a

contradiction, since the gain cross-section spectrum can be obtained by multiplying the fluorescence emission spectrum by a factor of λ^5 and, therefore, the gain maximum is shifted to the long-wavelength region⁴³. Similarly, for the well-known Ti:Sapphire laser the fluorescence emission peak is at 750 nm, while the gain maximum is located near 800 nm⁴³." (middle of page 14).

Only the first of the deficiencies mentioned above requires a major change to the manuscript. I consider the other ones as minor deficiencies. Once the raised issues are addressed (either by new experiments or thorough explanations), I am convinced that this manuscript will fully merit publication in Nature Communications.

Further suggestions to improve the manuscript.

Below, I have listed other comments that may help to improve the manuscripts (please use them as mere suggestions, no need to address them) and help to understand the underlying physics.

I. The potential significance of the result is high. However, more motivation and perspective would generally help the general reader to understand this significance better.

We add some important potential application of NV- laser in introduction.

II. Describe to the general reader what inhibited the development of defect-based lasers in diamond so far. The idea is very old, but there is no information on what were the major obstacles. How important is the use of the custom-made pulsed Nd:YAG laser? Are there any comparable commercial devices? The link between active mode-locking, possibility of controlled uniform (temporal?) pumping of the sample and observation of lasing is unclear to me.

There is a short remark in the text revealing the advantages of custom-made our laser.

"Note, that conventional Q-switched nanosecond Nd:YAG lasers usually have a stochastic structure of resonator light field time profile, which leads to significant pulse-to-pulse intensity variations with cavity round-trip period features. Such pump pulses cannot allow a controlled uniform pumping of the sample. In opposite, the train of active mode-locked picosecond laser pulses employed in our experiments has high temporal regularity and shot-to-shot repeatability³⁷." (bottom of page 9)

III. Fit the data on Fig. 5 with exponential functions and plot the amplitude and decay constant vs. pulse energy – I think that this could help to extract and visualize the essential physical information from these figures. I believe this might be nontrivial and in case of panel (b) and (c) and rate-equations might be necessary to model the data.

In the absence of induced absorption, the relaxation of stimulated emission has a simple exponential form. However, in the presence of induced absorption, the kinetics of the probe signal have a complex nonlinear dependence that cannot be reduced to one or two exponential processes. We hope that on the basis of the results of our work, theoretical modeling of these processes will be done in our or another laboratory. We believe that in this work, which is directly devoted to the experimental evidence of lasing, such modeling is redundant.

IV. Energetically, the lasing (around 720 nm, that is 1.72 meV) and pumping (2.33 meV) occur at a roughly comparable energetic distance from the zero phonon line (1.95 meV). This means

that the emission and absorption are assisted by the excitation of a vibrons of comparable energies. Do authors think that this could be anyhow relevant?

We believe that close coincidence of energies does not play any significant role. Our experiments on femtosecond luminescence upconversion in NV⁻ centers (unpublished yet) show that only the optical density depends on the pump wavelength. The luminescence signal in the phonon wing appears at all wavelengths simultaneously immediately after exposure to pumping (less than 10 fs) and is a simple convolution of the excitation pulse shape with an exponential function corresponding to the lifetime of the NV⁻ center. Moreover, in the absence of photoionization, no spectral kinetics is observed.

V. There are a couple of typos, like:

***which many orders above the CW lasers => which is many orders above the CW lasers
Nothing major or excessive, a careful check and using spell-checker should help.***

In the new version of the manuscript, we tried to correct all typos and inaccuracies.

Alexander Savvin, Alexander Dormidonov, Evgeniya Smetanina, Vladimir Mitrokhin, Evgeniy Lipatov, Dmitriy Genin, Sergey Potanin, Alexander Yelisseyev, Viktor Vins

REVIEWERS' COMMENTS

Reviewer #1 (Remarks to the Author):

The authors have addressed all queries and now the manuscript may be considered for publication.
Thank You
Referee

Reviewer #2 (Remarks to the Author):

We think that the authors have addressed most of the criticism raised in the first round of reviews in a sufficient manner. Once the additional comments raised below are addressed, we would be happy to see the paper printed in Nature Communications.

1) The authors have addressed the issues raised around Ref [27] ('indirect detection of stimulated emission') and have also added an additional reference (Ref [32]) and the following text:
"A rize of stimulated emission at wavelengths in the PSB of NV– zero-phonon line (ZPL) was detected by measuring a decrease of the NV– spontaneous emission under a CW optical pump at 532 nm. In the recent work 32 the amplification of 721-nm laser diod radiation by stimulated emission from NV– centres in diamond inside an optical fiber cavity was demonstrated. However a lasing at NV– centers in diamont have not been delivered yet." (bottom of page 2)

We highlight again that the work in Ref [27] actually detected stimulated emission through lock-in detection consistent with a simultaneous reduction in spontaneous emission. We encourage the authors to correct/extent their statement accordingly. Note that Ref [32] actually used a Ti:Sapphire laser and so the word 'diode' can be removed.

2) We still feel that the present paper falls significantly short of motivating the importance of NV lasing in the context of laser threshold magnetometry (Ref [20]). Again, given the importance of the findings reported here, a better motivation in this direction could be reasonably expected (e.g. instead of simply adding a reference, one sentence for providing more context might be interesting and relevant for the reader).

3) We would like to insist on the fact that the title of the present paper should indicate the pulsed nature of the NV lasing observed here. Given the strong interest and the still outstanding goal of achieving a CW laser, this would be appropriate. We agree though that no timescale of the pulses is necessary in the title.

4) We think that the manuscript has greatly gained by the addition of Figure 7a. Seeing this figure, we are wondering whether it would also be worth extracting the linewidth information from Figure 7b and plotting it alongside the data in Figure 7a (second y-axis). This would enable the reader to see the mentioned linewidth narrowing straight away and relate it directly to the laser threshold curve. We acknowledge that we did not mention this in round 1 of the review but seeing now Figure 7a, adding the linewidth information is an obvious and easy-to-do add-on that again would improve the readability of the paper.

5) We encourage the authors again to go through the manuscript with a fine-tooth comb to check again for typos, especially in the newly added paragraphs (e.g. 'rize', 'diod', and 'diamont' are just three examples of typos)

Reviewer #3 (Remarks to the Author):

My concerns have been satisfactorily addressed in the revised manuscript. I recommend publishing the article in Nature Communications.

Dear Editors!

We are very grateful for the opportunity to publish our manuscript “NV⁻ Diamond Laser” NCOMMS-21-08760A in Nature Communications and will try to carefully correct the remaining remarks of the Reviewers and edit our manuscript to comply with Nature Communications policies and formatting requirements. Just below there is responding letter where we noting the changes that have been made in the text according to Reviewers comments.

Reviewer #1:

The authors have addressed all queries and now the manuscript may be considered for publication.

***Thank You
Referee***

We appreciate the Referee for reviewing time our manuscript and very helpful suggestions.

Reviewer #2:

We think that the authors have addressed most of the criticism raised in the first round of reviews in a sufficient manner. Once the additional comments raised below are addressed, we would be happy to see the paper printed in Nature Communications.

We thank the Referee for constructive and interesting discussion of our work. We provide a point-by-point response to the Referee's round 2 comments.

1) The authors have addressed the issues raised around Ref [27] ('indirect detection of stimulated emission') and have also added an additional reference (Ref [32]) and the following text:

“A rise of stimulated emission at wavelengths in the PSB of NV⁻ zero-phonon line (ZPL) was detected by measuring a decrease of the NV⁻ spontaneous emission under a CW optical pump at 532 nm. In the recent work 32 the amplification of 721-nm laser diode radiation by stimulated emission from NV⁻ centres in diamond inside an optical fiber cavity was demonstrated. However a lasing at NV⁻ centers in diamond have not been delivered yet.” (bottom of page 2) We highlight again that the work in Ref [27] actually detected stimulated emission through lock-in detection consistent with a simultaneous reduction in spontaneous emission. We encourage the authors to correct/extent their statement accordingly. Note that Ref [32] actually used a Ti:Sapphire laser and so the word 'diode' can be removed.

The description of the papers [28] and [32] was finally corrected accordingly with all Referee's comments. The last paragraph of the page 6 was updated:

“Spectroscopic characteristics of NV⁻ centers in diamond were investigated in²⁷; but, stimulated emission or lasing at NV centers in diamond have not been achieved in that work. Stimulated emission at wavelengths in the PSB of NV⁻ zero-phonon line (ZPL) was registered in²⁸ through lock-in detection consistent with a simultaneous reduction in spontaneous emission under a CW optical pump at 532 nm. In the recent work³² the amplification of 721-nm laser radiation by stimulated emission from NV⁻ centers in

diamond inside an optical fiber cavity was demonstrated. However, a lasing at NV⁻ centers in diamond have not been realized yet.” (bottom of page 2)

2) We still feel that the present paper falls significantly short of motivating the importance of NV lasing in the context of laser threshold magnetometry (Ref [20]). Again, given the importance of the findings reported here, a better motivation in this direction could be reasonably expected (e.g. instead of simply adding a reference, one sentence for providing more context might be interesting and relevant for the reader).

We agree with the Referee that laser threshold magnetometer is one of interesting and perspective application that requires CW NV⁻ diamond laser. The first paragraph of the page 2 was updated and expanded:

“Laser initialization, changing and reading the spin state of an NV⁻ electron allows to control the state of a quantum system and turn NV⁻ centers in diamond into a promising platform for optical quantum computing¹²⁻¹⁹, quantum metrology, sensing and visualization, fluorescent magnetometry and laser threshold magnetometry that requires a CW laser constructed from the NV⁻ centers²⁰⁻²⁶.” (middle of page 2)

3) We would like to insist on the fact that the title of the present paper should indicate the pulsed nature of the NV lasing observed here. Given the strong interest and the still outstanding goal of achieving a CW laser, this would be appropriate. We agree though that no timescale of the pulses is necessary in the title.

The development of a CW laser on NV⁻ color centers in diamond is of strong interest for a wide range of possible applications and is one of the main goals of our current works. We agree with the Referee that the laser presented in our manuscript operates only in the pulsed pumping mode and CW generation mode has not been achieved. However, we have experimentally demonstrated lasing on NV⁻ color centers in diamond for the first time in spite of the continued efforts of other world scientific groups. In order to underline the fact of the realization of the NV⁻ laser with the permission of the Referee we would like not to reflect the laser operation mode in the title of the manuscript. We hope that our work could be the first step for the development of CW and ultrashort laser systems.

4) We think that the manuscript has greatly gained by the addition of Figure 7a. Seeing this figure, we are wondering whether it would also be worth extracting the linewidth information from Figure 7b and plotting it alongside the data in Figure 7a (second y-axis). This would enable the reader to see the mentioned linewidth narrowing straight away and relate it directly to the laser threshold curve. We acknowledge that we did not mention this in round 1 of the review but seeing now Figure 7a, adding the linewidth information is an obvious and easy-to-do add-on that again would improve the readability of the paper.

The dependence of the generated pulse linewidth on the pump energy is shown in Figure A inside this response letter. Since pulse linewidth is almost constant (see Figure A) and doesn't narrow with the pump energy increasing we do not include the Figure A in our manuscript.

Gaussian-like spectrum linewidth of the laser pulses formed in the optical cavity is inversely proportional to the expression $\sqrt{N(E_p)M}$, where M is the number of pulse passes through the active medium, and $N(E_p)$ is the inverse population density, which

increases with the pump energy (power) growth. Therefore, in many cases the lasing spectrum actually narrows with the optical pump energy increasing.

However, for the presented in our manuscript NV⁻ diamond laser the expression $\sqrt{N(E_p)M}$ (Figure B in this letter) remains almost constant with the pump pulses energy E_p increasing, since the time of the pulse output (see Figure 7b) and, accordingly, the number of resonator roundtrip M is reduced. The red dotted line in Figure A is a theoretical calculation of the laser pulse linewidth in accordance with the dependence $1/\sqrt{N(E_p)M}$. Since the linewidth dependence on the pump energy is almost constant and uninformative for readers, we do not include Figures A and B in our manuscript.

Figure A

Figure B

5) We encourage the authors again to go through the manuscript with a fine-tooth comb to check again for typos, especially in the newly added paragraphs (e.g. ‘rize’, ‘diod’, and ‘diamont’ are just three examples of typos).

All manuscript was updated and corrected.

Reviewer #3:

My concerns have been satisfactorily addressed in the revised manuscript. I recommend publishing the article in Nature Communications.

We appreciate the Referee for reviewing time our manuscript and very helpful suggestions.

Alexander Savvin, Alexander Dormidonov, Evgeniya Smetanina, Vladimir Mitrokhin, Evgeniy Lipatov, Dmitriy Genin, Sergey Potanin, Alexander Yelisseyev, Viktor Vins